# Phase Angle, a Cornerstone of Outcome in Head and Neck Cancer

**DOI:** 10.3390/nu14153030

**Published:** 2022-07-24

**Authors:** Daniel Sat-Muñoz, Brenda-Eugenia Martínez-Herrera, Javier-Andrés González-Rodríguez, Leonardo-Xicotencatl Gutiérrez-Rodríguez, Benjamín Trujillo-Hernández, Luis-Aarón Quiroga-Morales, Aldo-Antonio Alcaráz-Wong, Carlos Dávalos-Cobián, Alejandro Solórzano-Meléndez, Juan-Daniel Flores-Carlos, Benjamín Rubio-Jurado, Mario Salazar-Páramo, Gabriela-Guadalupe Carrillo-Nuñez, Eduardo Gómez-Sánchez, Arnulfo-Hernán Nava-Zavala, Luz-Ma-Adriana Balderas-Peña

**Affiliations:** 1Departamento de Morfología, Centro Universitario de Ciencias de la Salud (CUCS), Universidad de Guadalajara (UdG), 950 Sierra Mojada, Puerta 7, Edificio C, 1er Nivel, Guadalajara 44340, Jalisco, Mexico; 2Cuerpo Académico UDG CA-874 “Ciencias Morfológicas en el Diagnóstico y Tratamiento de la Enfermedad”, 950 Sierra Mojada, Puerta 7, Edificio C, 1er Nivel, Guadalajara 44340, Jalisco, Mexico; eduardo.gsanchez@academicos.udg.mx; 3Departamento Clínico de Oncología Quirúrgica, División de Oncología Hematología, Unidad Médica de Alta Especialidad (UMAE), Hospital de Especialidades (HE), Centro Médico Nacional de Occidente (CMNO), Instituto Mexicano del Seguro Social (IMSS), 1000 Belisario Domínguez, Guadalajara 44340, Jalisco, Mexico; 4Comité de Cabeza y Cuello, UMAE, Hospital de Especialidades, Centro Médico Nacional de Occidente, Instituto Mexicano del Seguro Social, 1000 Belisario Domínguez, Guadalajara 44340, Jalisco, Mexico; aldo.alcaraz@imss.gob.mx (A.-A.A.-W.); dr_solorzano@yahoo.com.mx (A.S.-M.); 5Hospital General de Zona (HGZ), #02 c/MF “Dr. Francisco Padrón Puyou”, Órgano de Operación Administrativa Desconcentrada San Luis Potosi, IMSS, San Luis Potosi 78250, San Luis Potosi, Mexico; bren.mtzh16@gmail.com; 6Unidad de Investigación Biomédica 02, UMAE Hospital de Especialidades (HE), Centro Médico Nacional de Occidente (CMNO), Instituto Mexicano del Seguro Social (IMSS), 1000 Belisario Domínguez, Guadalajara 44340, Jalisco, Mexico; javier.gonzalez5722@alumnos.udg.mx (J.-A.G.-R.); xicotencatl.gutierrez@alumno.udg.mx (L.-X.G.-R.); luisquiroga@hotmail.com (L.-A.Q.-M.); navazava@yahoo.com.mx (A.-H.N.-Z.); 7Doctorado en Ciencias Médicas, Facultad de Medicina, Universidad de Colima, Colima 28040, Colima, Mexico; trujillobenjamin@hotmail.com; 8Carrera de Médico Cirujano y Partero, Coordinación de Servicio Social, Centro Universitario de Ciencias de la Salud (CUCS), Universidad de Guadalajara (UdG), Guadalajara 44340, Jalisco, Mexico; 9Carrera de Médico Cirujano y Partero, Coordinación de Servicio Social, Centro Universitario del Sur, Universidad de Guadalajara (UdG), Ciudad Guzmán 49000, Jalisco, Mexico; 10Comisión Interinstitucional de Formación de Recursos Humanos en Salud, Programa Nacional de Servicio Social en Investigación 2021, Demarcación Territorial Miguel Hidalgo 11410, Ciudad de México, Mexico Programa de Doctorado en Investigaciȯn Clínica, Coordinación de Posgrado, Centro Universitario de Ciencias de la Salud (CUCS), Universidad de Guadalajara (UdG), Guadalajara 44340, Jalisco, Mexico; 11Unidad Académica de Ciencias de la Salud, Consultor Nutricional en la Clínica de Rehabilitación y Alto Rendimiento ESPORTIVA, Universidad Autónoma de Guadalajara, Zapopan 45129, Jalisco, Mexico; 12Departamento Clínico de Anatomía Patológica, División de Diagnóstico, UMAE, Hospital de Especialidades, Centro Médico Nacional de Occidente, Instituto Mexicano del Seguro Social, 1000 Belisario Domínguez, Guadalajara 44340, Jalisco, Mexico; 13Departamento Clínico de Gastroenterología, Servicio de Endoscopía, División de Medicina, UMAE, Hospital de Especialidades, Centro Médico Nacional de Occidente, Instituto Mexicano del Seguro Social, 1000 Belisario Domínguez, Guadalajara 44340, Jalisco, Mexico; endosmedica@gmail.com; 14Departamento Clínico de Oncología Radioterapia, Servicio Nacional de Radioneurocirugía, División de Oncología Hematología, UMAE, Hospital de Especialidades, Centro Médico Nacional de Occidente, Instituto Mexicano del Seguro Social, 1000 Belisario Domínguez, Guadalajara 44340, Jalisco, Mexico; 15Departamento Clínico de Cirugía General, Servicio de Soporte Nutricio, División de Cirugía, UMAE, Hospital de Especialidades, Centro Médico Nacional de Occidente, Instituto Mexicano del Seguro Social, 1000 Belisario Domínguez, Guadalajara 44340, Jalisco, Mexico; dr_danielflores@hotmail.com; 16Departamento Clínico de Hematología, División de Oncología Hematología, UMAE, Hospital de Especialidades, Centro Médico Nacional de Occidente, Instituto Mexicano del Seguro Social, 1000 Belisario Domínguez, Guadalajara 44340, Jalisco, Mexico; rubiojuradob@gmail.com; 17Academia de Inmunología, Departamento de Fisiología, Centro Universitario de Ciencias de la Salud (CUCS), Universidad de Guadalajara (UdG), 950 Sierra Mojada, Gate 7, Building O, 1st Level, Guadalajara 44340, Jalisco, Mexico; mario.sparamo@academicos.udg.mx; 18Departamento de Microbiología y Patología, Cuerpo Académico CAC 365 Educación y Salud, Centro Universitario de Ciencias de la Salud (CUCS), Universidad de Guadalajara (UdG), 950 Sierra Mojada, Gate 7, Building O, 1st Level, Guadalajara 44340, Jalisco, Mexico; gabriela.cnunez@academicos.udg.mx; 19División de Disciplinas Clínicas, Centro Universitario de Ciencias de la Salud (CUCS), Universidad de Guadalajara (UdG), 950 Sierra Mojada, Edificio N, Puerta 1, Planta Baja, Guadalajara 44340, Jalisco, Mexico; 20Unidad de Investigación Social Epidemiológica y en Servicios de Salud, Órgano de Operación Administrativa Desconcentrada, Guadalajara 44340, Jalisco, Mexico; 21Programa Internacional Facultad de Medicina, Universidad Autónoma de Guadalajara, Av. Patria 1201, Lomas del Valle, Zapopan 45129, Jalisco, Mexico; 22Servicio de Inmunología y Reumatología, División de Medicina Interna, Hospital General de Occidente, Secretaria de Salud Jalisco, Av. Zoquipan 1050, Zapopan 45170, Jalisco, Mexico

**Keywords:** phase angle, head and neck cancer (H&NC), survival, risk of death

## Abstract

In patients with head and neck cancer, malnutrition is common. Most cases are treated by chemo-radiotherapy and surgery, with adverse effects on the aerodigestive area. Clinical and biochemical characteristics, health-related quality of life, survival, and risk of death were studied. The selected subjects were divided into normal- and low-phase-angle (PA) groups and followed up for at least two years. Mean ages were 67.2 and 59.3 years for low and normal PA, respectively. Patients with PA < 4.42° had significant differences in age, anthropometric and biochemical indicators of malnutrition, and inflammatory status compared to patients with PA > 4.42°. Statistical differences were found in the functional and symptom scales, with lower functional scores and higher symptom scores in patients with low PA. Median survival was 19.8 months for those with PA < 4.42° versus 34.4 months for those with PA > 4.42° (*p* < 0.001).The relative risk of death was related to low PA (2.6; *p* < 0.001). The percentage of living patients (41.7%) is almost the same as the percentage of deceased subjects (43.1%; *p* = 0.002), with high death rates in patients with PA < 4.42°. Phase angle was the most crucial predictor of survival and a risk factor for death in the studied cases.

## 1. Introduction

In patients with head and neck cancer (H&NC), malnutrition is a frequent condition even before treatment, regardless of the anatomical site of the primary tumor (nasal, oral, hypopharynx, and larynx). Most of them are treated by chemo-radiotherapy and surgery; however, these are usually associated with adverse events such as dysphagia, mucositis, nausea, and other aerodigestive symptoms [1,2,3].

The symptoms associated with the high metabolic rate of this type of tumor contribute to weight loss linked to malnutrition [4], loss of function, and a high rate of adverse outcomes, even with the best treatment, with a high rate of morbidity from complications and mortality [5,6].

Different resources report that in newly diagnosed patients who are naïve to treatment, 3 to 52% show some degree of malnutrition, and during treatment, the percentage increased to 44 to 88% [7].

Bioelectrical impedance analysis (BIA) is a crucial tool used in the clinical setting to assess body composition in different types of patients and allows analysis of not only weight loss but also changes in muscle mass and, through phase angle (PA), the integrity of membranes in cells [8,9,10,11]. These are related to functionality, quality of life, and even the risk of complications and mortality [12].

BIA measures the resistance and capacitance of the body and its various tissues by recording a voltage drop across an applied current. The capacitance generates a delay in the voltage relative to the current. The geometric representation of the delay is called the phase angle and represents the angular transformation of the relationship between capacitance and resistance [11,12,13].

The phase angle represents the resistance generated by body fluids and cell membranes (capacitance) in the human body. It is positively correlated with capacitance and negatively correlated with resistance [14,15]. Healthy people have a phase angle between four and ten degrees [11]. A low phase angle value is linked/related to cell death or damage to the integrity of cell membranes, and a medium or high value refers to intact cells and the expected amount of water ratio between intracellular and extracellular spaces, which predicts the cell mass of the body [14,15].

Cancer patients, particularly head and neck cancer patients, suffer from malnutrition and loss of lean mass before, during, and after treatment [16]. Therefore, we consider that the phase angle could be a cornerstone for predicting outcome, functionality, response to treatment, and mortality in this specific group of patients.

This report aimed to determine the role of phase angle in the outcomes of head and neck cancer patients in a population with a high prevalence of overweight and obesity.

## 2. Materials and Methods

This prospective cohort analysis of naïve patients with head and neck cancer followed all of them for at least two years. It was reviewed and approved by the Institutional Review Board of the Instituto Mexicano del Seguro Social 1301. All patients signed an informed consent to participate in the study. The procedures were carried out according to the standards considered in the Declaration of Helsinki.

The EORTC-validated questionnaires for Mexican Spanish QLC-C30 and H&N35 were used to assess health-related quality of life (HRQoL). In total, 120 patients were recruited to form a consecutive non-probability sample, aged 30–85 years, with biopsy-confirmed head and neck cancer before initiating multidisciplinary treatment in a tertiary hospital in Guadalajara, México.

Patients with two or more malignant neoplasms, autoimmune illness, chronic lung or renal disease, or any contraindication to perform bioelectrical impedance analysis (metal prostheses or electronic implant devices, limb amputations, or severe edema) were excluded. Patient demographics, clinical stage, anatomical location of the tumor, treatment modality, and laboratory tests were extracted from patient records.

A dietitian validated the body composition analysis and classified the phenotype of the patients as sarcopenia, sarcopenic obesity, obesity, or typical body composition. Before the BIA, patients fasted for 8 h and measurements were performed without jewelry or socks. The SECA 213 height scale (Seca, Germany) was used to measure height. The mBCA SECA 514 bioelectric impedance device (Seca, Germany) determined patients’ weight, phase angle, total skeletal muscle mass, and total body fat percentage, with data used to calculate the body mass index (BMI) and skeletal muscle mass index (SMMI). All of these were used to calculate patient outcomes.

BMI was calculated as described by the World Health Organization. We obtained the SMMI by dividing the total skeletal muscle mass (kg) by the height squared (m^2^). Based on both measurements, patients were classified into three study groups: (1) non-sarcopenic group (NSG): women SMMI ≥ 6.42 kg/m^2^, men ≥ 8.86 kg/m^2^, and BMI < 25 kg/m^2^; (2) sarcopenic group (SG): female SMMI < 6.42 kg/m^2^, male < 8.87 kg/m^2^, and a BMI < 25 kg/m^2^); and (3) sarcopenic obesity group (SOG): women SMMI < 6.42 kg/m^2^, men < 8.87 kg/m^2^, and a BMI ≥ 25 kg/m^2^ [17,18].

The ratio of SMMI to patient weight was used to calculate the proportion of total weight corresponding to muscle mass, normalized by dividing by the square of height. Hand-grip strength was assessed using a Jamar Plus+ Digital hand dynamometer (Patterson Medical Supply, Cedarburg, WI, USA). According to the American Association of Hand Therapists, patients held the device and compressed it with maximum force to obtain a maximum contraction. The test was repeated three times for each hand, with one-minute rest intervals between measurements. The highest result of all tests was recorded [19].

The EORTC QLQ-C30 v.3, (Brussels, Belgium) questionnaire (validated for Mexican Spanish) was conducted to assess HRQoL. The instrument consists of six multi-item scales (related to patient functioning) and nine single-item scales (describing the severity of cancer-related symptoms). The EORTC QLQ-H&N35 supplementary module for H&NC patients was also used. It consists of 35 questions, 7 multi-item symptom scales, and 11 single-item symptom scales described by the EORTC Scoring Manual [20].

Analysis of the two items of the EORTC questionnaire required linear transformation of each item or multi-item scale to obtain a range of scores from 0 to 100. Higher scores on the functioning scales indicate that patients perceive a better QoL than lower scores. Conversely, higher scores on the symptoms scales indicate a higher frequency of severe symptoms and thus a worse QoL [20].

Statistical analysis was performed with the software package IBM^®^ SPSS^®^ Statistics version 28 (Armonk, NY ^®^, USA).

Results are presented as means ± standard deviation (SD) for variables with normal distributions. Non-parametric variables were described as medians (interquartile intervals (IQIs)).

Categorical variables were expressed as numbers and percentages of the total. Pearson’s chi-square tests were performed to assess differences between the two groups (Fishers’ tests if the estimated values were <5), and one-way ANOVA and Kruskal–Wallis tests with Bonferroni correction were used to assess differences between the three groups. To determine the relationship, Pearson’s correlation or Spearman’s Rho was calculated depending on the type of variable. Survival analysis was carried out using the Kaplan–Meier method and hazard risk was estimated by Cox regression. Analyses were two-sided, and a *p*-value < 0.05 was considered significant. Cronbach’s alpha value was used for reliability in the multi-item scales of the EORTC questionnaires.

## 3. Results

In total, 139 patients with H&NC with a mean age of 63.5 (±13) years, including 32 (23%) women and 107 (77%) men, were studied.

In total, 49 patients without sarcopenia, 52 with sarcopenia, and 38 with sarcopenia obesity were found. The mean phase angle was 4.29° (±0.93; mean CI95% = 4.14° to 4.95°) for the 139 studied patients. According to the sarcopenia phenotype, all three groups showed significant differences between non-sarcopenia and sarcopenia (4.7° (±0.83) versus 4.0° (±0.91); *p* < 0.001) and non-sarcopenia and sarcopenic-obesity (4.7° (±0.83) versus 4.18° (±0.84); *p* = 0.023).

Based on the above findings, it was decided to set the cut-off point of the normal phase angle at 4.42°, which coincides with the Youden index of the ROC curve (cut-off point, with the Kolmogorov–Smirnoff (K-S) maximum corresponding to a phase angle of 4.42° and an overall model quality score of 0.55 (sensitivity 0.621; 1-specificity 0.36).

### 3.1. Characteristics of the Cohort of Head and Neck Cancer Patients

#### 3.1.1. Clinical Aspects

Table 1 describes the results according to a normal (4.42°) or low (<4.42°) phase angle for the studied group of H&NC patients and the clinical characteristic of the studied patients. The table shows that male gender predominates over female (*p* = 0.035), and the percentage of living patients (41.7%) is almost the same with respect to dead subjects (43.1%; *p* = 0.002), with high death rates in patients with a phase angle <4.42°.

The predominant anatomical locations of H&NC were the larynx and oral cavity, accounting for 69.7% (*n* = 97) of the cases studied. Squamous/epidermoid histology was predominant (*n* = 123; 88%). Clinical-stage IV accounted 45% of the cases (*n* = 63), and 64.7% (*n* = 90) of the patients had sarcopenia or sarcopenic obesity.

#### 3.1.2. Anthropometrical and Biochemical Indicators

The anthropometric and biochemical parameters revealed significant differences between patients with a low phase angle and those with a normal phase angle in the following parameters: age; hand grip strength; gait speed; BMI (*p* = 0.042), with a lower BMI but a higher fat percentage in the cases with a lower phase angle; SMMI; hemoglobin; absolute lymphocyte count; serum albumin; and C-reactive protein; all the afore-mentioned parameters are malnutrition and inflammatory indicators (Table 2).

### 3.2. Characteristics of the Health-Related Quality of Life in the Head and Neck Cancer Patient Cohort

#### 3.2.1. Results of the EORTC QLQ-C30 Questionnaire

Regarding the health-related quality of life functioning and symptom scales, statistical differences were found in the following scores: global health status/quality of life, physic functioning, role functioning, fatigue, pain, insomnia, and loss of appetite for the EORTC QLQ-C30 questionnaire, with lower scores on the functional scales and higher scores on symptoms in patients with a low phase angle (Table 3).

#### 3.2.2. Results of the EORTC QLQ-H&N35 Questionnaire

The EORTC QLQ-H&N35 scales based on the specific symptoms of H&NC patients showed significantly higher scores in patients with a low phase angle for the following items: pain, swallowing, senses problems, trouble with social eating, teeth, opening mouth, dry mouth, sticky saliva, felt ill, pain killers, and nutritional supplements (Table 3).

### 3.3. Survival Analysis of the Head and Neck Cancer Patient Cohort

#### Kaplan-Meier and Cox Regression Analysis Results

The percentage of death patients reach 70% in patients with low phase angle and it is 30% in subjects with preserved phase angle (Table 4). Median survival was 19.8 months (95% confidence interval (CI), 15.6 to 24.1 months) in the phase angle <4.42 group (*n* = 74) and 34.4 months (95% CI, 29.6 to 39.2) in the comparison group (*n* = 65; phase angle ≥ 4.42: relative risk by Cox’s regression for death, hazard ratio (HR) = 2.6, 95%CI = 1.52 to 4.6; *p* < 0.001, Figure 1).

Using the sarcopenia/sarcopenic obesity to classify the study subjects, the mean survival was 24 months (95%CI= 23.1 to 31 months) in sarcopenic subjects (*n* = 90). In the non-sarcopenic group (*n* = 49), the mean survival was 34.2 months (95%CI = 28.1 to 40.3; *p* = 0.033).

## 4. Discussion

Body composition was measured by BIA (bioelectrical impedance analysis), which is a non-invasive, inexpensive, and relatively easy method to perform [21]. PA is a parameter obtained through reactance and resistance [13]. Resistance represents the opposition of a structure or tissue to electrical current through the body, and reactance is the resistance presented by the cell membrane and its lipid bilayer with amphipathic characteristics [13,22].

PA is, therefore, a parameter used for the assessment of both the integrity of cell membranes throughout the body and nutritional status. It has been proposed as a prognostic factor in chronic diseases associated with frailty, physical wear and tear, and acute diseases with a direct influence on the protein reserves of the body [11], such as oncological diseases [23].

PA shows some physiological variations, such as lower values in women than in men, lower values after the age of 60, and changes in different racial groups, with higher values in African descent and Caucasian populations and lower values in Asian and Latino populations [11,24,25,26]. In our current observations, we found significant differences between genders when estimating the percentage of individuals with low PA (Table 1), with more than 68% of women affected. 

Low PA values in chronic diseases and cancer are associated with an adverse prognosis, prolonged hospital stay, and worse survival than normal values [25,27,28,29,30,31]. We found that the same situation is present in the number of deceased subjects with low PA (70%), oral cavity involvement and low PA (69.8%), or the presence of low PA in terminally ill patients (CS lV 63.5%) and the proportion of sarcopenia/sarcopenic obesity among patients related to low PA (63%), coinciding with the findings of Do-Amaral-Paes [30] and Leon-Idougurram [32] in 2018 and 2022, respectively. Our current results reveal the relationship between PA, clinical, biochemistry, PRO (patient-reported outcomes), and survival/death risk in Mexican H&NC subjects with a high prevalence of overweight-obesity [33,34].

Accurate nutritional diagnosis could allow early detection of nutritional risks prior to worsening of HRQoL secondary to complications and death related to impaired nutritional status [29,31,32,33].

Our results shown in Table 2 coincide with those of the report by León-Idougurram and Do-Amaral-Paes, where we found alterations in the anthropometric and nutritional parameters and inflammatory markers in all three groups at the expense of subjects with low PA [30,32]. However, in 2020, some authors, such as Yasui-Yamada, considered the clinical nutritional significance of PA to be ambiguous [24].

It is essential to highlight the role of tumor cachexia and inflammatory mediators such as C-reactive protein, with a significant increase in patients with PA < 4.42° and sarcopenia, and its relationship with a decreased hand grip strength and gait speed and its negative impact on functional markers in this group of patients. Other altered biochemical indicators reflecting malnutrition and its deleterious cellular effect are low hemoglobin, absolute lymphocyte, and albumin levels in blood patients with low PA, as recently described by the Leon-Idougurram group [32].

The evaluation of QoL of patients with H&NC based on the EORTC QLQ-C30 and QLQ-H&N35 instruments is widely used worldwide. One of the first reports demonstrating the reliability and validity of these was published by Sherman AC in 2000 [35], who analyzed a group of 120 outpatients with advanced H&NC in several clinical scenarios: (1) pre-treatment, (2) during the active phase of treatment, (3) 6-month follow-up, and (4) follow-up over 6 months, compared to patients without malignant disease. Their observations provided insight into the psychometric behavior of the instruments and support the results of the subsequent HRQoL assessment with the same questionnaires, whereby we achieved valid psychometric results for the scales used in both instruments.

For the current cohort, we highlighted the impact of a low PA on global health status, and on functional scales such as physical functioning, role functioning, and symptoms including fatigue, pain, insomnia, and loss of appetite, within the EORTC QLQ-C30 (Table 3).

As previously described, malnutrition affects 50% of patients with H&NC [36] and more than 20% are classified as severe [32,37], leading to high rates of treatment complications and poor response, increasing the risk of delayed wound healing, infections, and deleterious effect on HRQoL [32,33,38]. In the specific EORTC QLQ-H&N35 module, the scales mainly affected are pain, swallowing, sense problem, trouble with social eating, teeth, opening mouth, dry mouth, sticky saliva, felt ill, pain killers, and nutritional supplements, all with higher scores for the low-PA group than the normal-PA group.

The above results allow us to infer that patient-perceived health status (patient-reported outcome) is closely related to measures of functioning such as gait speed and handgrip strength, in parallel to measures of body composition, specifically phase angle. 

Davide de Cicco et al., in 2021 [39], observed that HRQoL is severely impaired in subjects with oral cancer. They studied and described how HRQoL became an established practice worldwide, and has a close relationship with survival, satisfaction, and self-perception, considering it as a secondary endpoint of medical interventions in oncology patients.

The group from Sweden led by Axelsson, in 2018 [40], published the value of the phase angle as a prognostic factor in 128 subjects with H&NC, using a slightly higher cut-off than the one used in our current report (5.91° versus 4.42° used by us). These patients did not have a significant risk of death associated with gender, but age over 60 years accounted for a hazard ratio (HR) of 1.030 (95%CI: 1–1.062; *p* = 0.05), performance status as a reflection of functional status below 80 showed an HR of 5.4 (95%CI: 2.77–10.54; *p* < 0.001), weight loss percentage with an HR of 1.06 (95%CI 1.025–1.107; *p* = 0.001), and a normal phase angle represented a risk factor reduction HR:0.47 (95%CI: 0.36–0.62; *p* < 0.001), which are results that are similar to our findings. However, our population had a higher percentage of malnutrition, resulting in worse outcomes with a short follow-up (2 years), compared to their 12-year follow-up.

## 5. Conclusions

The phase angle in this cohort had a lower cut off than that reported by other groups, reflecting the prevalence of malnutrition in our population, which negatively impacts the HRQoL, outcome, and mortality percentage during the first two years of follow-up, and a significant increase in the HR for death. In summary, it is essential in areas with a high prevalence of obesity, where a multidisciplinary and transdisciplinary treatment team need to take actions directed toward the early detection of patients with a high malnourishment risk and perform intense nutritional management to avoid or limit sarcopenia, sarcopenic obesity, or tumor cachexia. 

## Figures and Tables

**Figure 1 nutrients-14-03030-f001:**
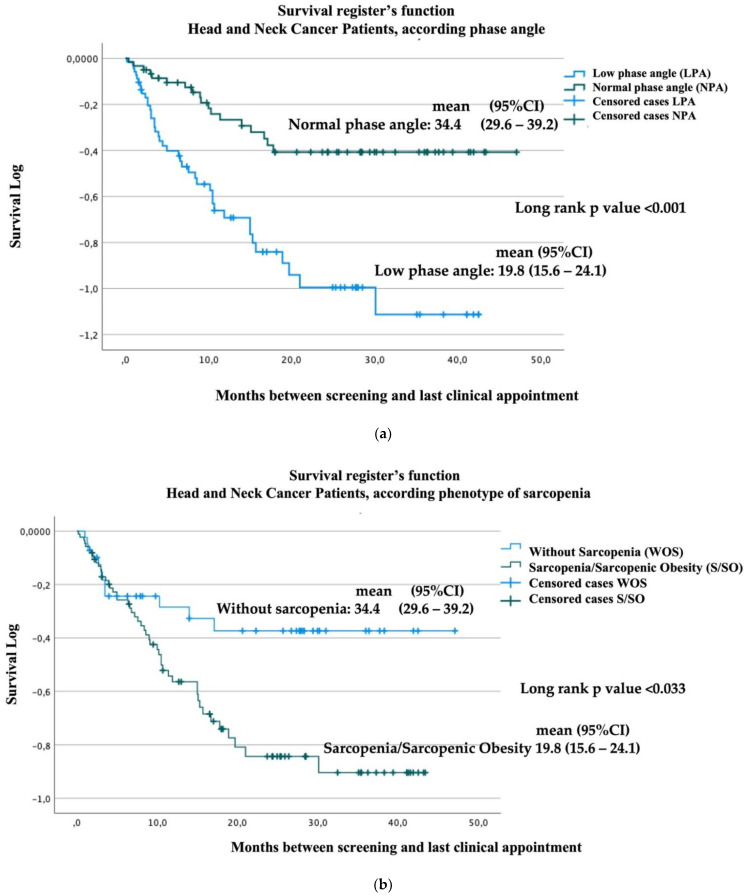
Kaplan–Meier survival analysis: (**a**) Log survival function by phase angle; (**b**) log survival function by sarcopenia phenotype.

**Table 1 nutrients-14-03030-t001:** Characteristic of the cohort of head and neck cancer patients studied.

Clinical Characteristic	Phase Angle < 4.42° *n* (% Inside Specific Group)	Phase Angle ≥ 4.42° *n* (% Inside Specific Group)	Total *n* (% Total Patients)	*p* Value *
**Gender**				
Female	22 (68.8%)	10 (31.3%)	32 (100%)	
Male	52 (48.6%)	55 (51.4%)	107 (100%)	
Total	74 (53.2%)	65 (46.8%)	139 (100%)	0.035
**Anatomical Location**				
Oral cavity	30 (69.8%)	13 (30.2%)	43 (100%)	
Larynx	23 (42.6%)	31 (57.4%)	54 (100%)	
Pharynx	3 (30%)	7 (70%)	10 (100%)	
Salivary glands	3 (50%)	3 (50%)	6 (100%)	
Nose	8 (53.3%)	7 (46.6%)	15 (100%)	
Skin	5 (62.5%)	3 (37.5%)	8 (100%)	
Unknown	1 (100%)	0 (0%)	1 (100%)	
Others	1 (50%)	1 (50%)	2 (100%)	
Total	74 (53.2%)	65 (46.8%)	139 (100%)	0.160
**Histology**				
Squamous or epidermoid	65 (52.8%)	58 (47.2%)	123 (100%)	
Others	9 (56.3%)	7 (43.7%)	16 (100%)	
Total	74 (53.2%)	65 (46.8)	139 (100%)	0.506
**Clinical Stage**				
I	10 (45.5%)	12 (54.5%)	22 (100%)	
II	8 (57.1%)	6 (42.9%)	14 (100%)	
III	8 (32%)	17 (68%)	25 (100%)	
IV	40 (63.5%)	23 (36.5%)	63 (100%)	
Non classified	8 (53.3%)	7 (46.7%)	16 (100%)	
Total	74 (53.2%)	65 (46.8%)	139 (100%)	0.099
**Phenotype by Body Composition**				
No Sarcopenia	17 (34.7%)	32 (65.3%)	49 (100%)	
Sarcopenia	35 (67.3%)	17 (32.7%)	52 (100%)	
Sarcopenic obesity	22 (57.9%)	16 (42.1%)	38 (100%)	
Total	74 (53.2%)	65 (46.8%)	139 (100%)	0.004

* Significant *p* value < 0.05. Chi squared test.

**Table 2 nutrients-14-03030-t002:** Characteristic of the anthropometric and biochemical parameters in the cohort of head and neck cancer patients studied.

Anthropometrical and Biochemical Indicators	Phase Angle < 4.42° *n* = 74 Mean (SD Standard Deviation)	Phase Angle ≥ 4.42° *n* = 65 Mean (SD Standard Deviation)	*p* Value *
**Age and Anthropometrical Indicators**			
Age	67.2 (12.32)	59.3 (12.57)	0.001
Handgrip strength	22.6 (8.0)	31.1 (8.0)	<0.001
Gait speed	0.75 (0.2)	0.92 (0.2)	<0.001
Phase angle	3.6 (0.6)	5.1 (0.5)	0.001
Body Mass Index (BMI)	24.9 (6.0)	26.7 (4.2)	0.042
Total Fat percentage	33.5 (11.2)	30.7 (8.9)	<0.057
Skeletal Muscle Mass Index (SMMI)	7.3 (3.4)	10.1 (5.1)	<0.001
**Biochemical Indicators**			
Hemoglobin	12.6 (1.9)	13.9 (1.7)	<0.001
Absolute lymphocyte count	1587 (946)	1920 (1016)	0.024
Albumin **	4.1 (3.6–4.3)	4.4 (4.15–4.5)	<0.001
C-Reactive Protein **	23.15 (5.8–28.25)	10.4 (2.9–21.3)	<0.001
Total cholesterol	184.2 (50.0)	183.5 (32.3)	0.465

* Significant *p* value < 0.05. Student’s T test. ** Significant *p* value < 0.05. U de Mann–Whitney test. Nonparametric distribution values. Median (interquartile interval).

**Table 3 nutrients-14-03030-t003:** EORTC QLQ-C30 and EORTC QLQ-H&N35 in the studied head and neck cancer patient cohort.

Scores for the QLQ Scales	Phase Angle < 4.42° *n* = 74 Mean (SD Standard Deviation)	Phase Angle ≥ 4.42° *n* = 65 Mean (SD Standard Deviation)	*p* Value *
**EORTC QLQ-C30 (SCORE 0–100)**			
Global Health Status/Quality of Life	62.2 (25.7)	76.4 (21.7)	<0.001
Physic Functioning **	70 (40–93)	93.3 (80–100)	<0.001
Role Functioning	83.3 (33.3–100)	100 (75–100)	0.003
Emotional Functioning	67.7 (27.9)	72.4 (27.3)	0.314
Cognitive Functioning	79.9 (21.0)	83.9 (20.6)	0.264
Social Functioning	71.4 (30.5)	79.5 (33.0)	0.135
Fatigue	47.7 (29.0)	21.4 (25.47)	<0.001
Nausea and vomiting	8.3 (17.3)	10.8 (25.6)	0.508
Pain **	16.7 (0–50)	0.0 (0–33.3)	0.009
Dyspnea **	0.0 (0–33.3)	0.0 (0–33.3)	0.121
Insomnia	45.5 (39.2)	31.8 (37.9)	0.039
Loss of appetite **	16.7 (0.0–66.7)	0 (0.0–0.0)	<0.001
Constipation **	32 (36.4)	26.7 (35.5)	0.386
Diarrhea	6.3 (14.2)	6.1 (19.4)	0.958
Financial Difficulties	40.1 (35.7)	31.8 (35.1)	0.171
**EORTC QLQ-H&N35 (SCORE 0–100)**			
Pain **	33.3 (14.6–58.3)	16.7 (0–29.2)	0.002
Swallowing	32.4 (27.0)	20.3 (26.7)	0.009
Senses problems **	23.3 (0.0–54.2)	0.0 (0.0–30)	0.003
Speech problems	42.5 (31.4)	33.1 (31)	0.080
Trouble with social eating	30.5 (31)	19.3 (27.3)	0.026
Trouble with social contact **	6.7 (0.0–26.7)	0.0 (0.0–10.3)	0.057
Less sexuality	43.5 (13.4)	41 (11.2)	0.206
Teeth **	20.7 (0.0–66.7)	0.0 (0.0–33.3)	0.023
Opening mouth **	33.3 (0.0–100)	0.0 (0.0–28.1)	<0.001
Dry mouth	46.8 (38.4)	30.6 (32.5)	0.008
Sticky saliva **	37.3 (0.0–100)	0.0 (0.0–41.3)	0.001
Coughing	33.5 (34.7)	25.0 (30.0)	0.128
Felt ill **	29.1 (0.0–66.7)	0.0 (0.0–33.3)	0.023
Pain killers	68.1 (45.5)	45.0 (47.4)	0.004
Nutritional supplements **	100 (0.0–100)	0.0 (0.0–60.4)	0.002
Feeding tube **	0.0 (0.00–0.00)	0.0 (0.00–0.00)	0.225
Weight loss	56.0 (48.1)	50.0 (48.0)	0.462
Weight gain **	0.0 (0.00–0.00)	0.0 (0.0–4.5)	0.072

* Significant *p* value < 0.05. Student’s T test. ** Significant *p* value < 0.05. U de Mann–Whitney test. Nonparametric distribution values. Median (interquartile interval).

**Table 4 nutrients-14-03030-t004:** Survival, death, and loss of follow-up.

Survival Status				
Clinical Characteristic	Phase Angle < 4.42° *n* (% Inside Specific Group)	Phase Angle ≥ 4.42° *n* (% Inside Specific Group)	Total *n* (% Total Patients)	*p* Value *
Alive	22 (37.9%)	36 (62.1%)	58 (41.7%)	
Death	42 (70%)	18 (30%)	60 (43.1%)	
Lost to follow-up	10 (47.6%)	11 (52.4%)	21 (15.2%)	
Total	74 (53.2%)	65 (46.8%)	139 (100%)	0.002

* Significant *p* value < 0.05. Chi squared T test.

## Data Availability

The datasets generated and/or analyzed during the current study are not publicly available because they are the property of the Instituto Mexicano del Seguro Social. Institutional and federal dispositions restrict unlimited access to personal data, but they are available from the corresponding authors on reasonable request with prior authorization from the institution.

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
