# Peer review of "Phase Angle, a Cornerstone of Outcome in Head and Neck Cancer"

_nutrients, 2022, doi:10.3390/nu14153030_

Round 1
Reviewer 1 Report
A very interesting study showing the importance of phase angle as a predictor of survival and a risk factor for death in studied cases. Only minor queries before publication:
Conclusions should be expanded, also describing the possible future developments following this study's results.
Line 92 you should add: " the rate of malnutrition is even higher in metastatic patients" and cite doi: 10.3390/curroncol28040213. and doi: 10.3390/medicina57060563.
Thank You
Author Response
COVER LETTER REVIEWER 1
Open Review
(x) I would not like to sign my review report
( ) I would like to sign my review report
English language and style
( ) Extensive editing of English language and style required
( ) Moderate English changes required
(x) English language and style are fine/minor spell check required
( ) I don't feel qualified to judge about the English language and style
|
Yes |
Can be improved |
Must be improved |
Not applicable |
|
|
Does the introduction provide sufficient background and include all relevant references? |
( ) |
(x) |
( ) |
( ) |
|
Are all the cited references relevant to the research? |
( ) |
(x) |
( ) |
( ) |
|
Is the research design appropriate? |
( ) |
(x) |
( ) |
( ) |
|
Are the methods adequately described? |
( ) |
(x) |
( ) |
( ) |
|
Are the results clearly presented? |
( ) |
(x) |
( ) |
( ) |
|
Are the conclusions supported by the results? |
( ) |
(x) |
( ) |
( ) |
Comments and Suggestions for Authors
A very interesting study showing the importance of phase angle as a predictor of survival and a risk factor for death in studied cases. Only minor queries before publication:
Conclusions should be expanded, also describing the possible future developments following this study's results.
Line 92 you should add: " the rate of malnutrition is even higher in metastatic patients" and cite doi: 10.3390/curroncol28040213. and doi: 10.3390/medicina57060563.
Thank You
Submission Date
28 June 2022
Date of this review
11 Jul 2022 11:16:09
Response to reviewer 1
We have reviewed the two articles proposed for citation, and neither describes the rate of malnutrition in patients with metastatic disease, however both papers describe different treatment modalities, so we have decided to include the systematic review on line 95, without additional text to avoid detracting from the focus of the cohort.
Considering the second reviewer comments we realized changes in results and conclusion highlighted in yellow.
|
|
|
|
|
Dr. Luz Ma. Adriana Balderas Peña. M.D., PhD. Belisario Dominguez No. 1000. Colonia Independencia. Guadalajara, Jalisco, México, CP 44340 Telephone (office): +52 33 3668 3000 extension 31611 Telephone (mobile): +52 33 3115 7678 e-mail: luz.balderas@academicos.udg.mx, alternative e-mail: lmabp@yahoo.com.mx |
|
Dr. Daniel Sat-Muñoz MD, MBA 950 Sierra Mojada, Gate 7, Building C, 1st level, Colonia Independencia, Guadalajara, Jalisco 44340, México +52 33 3668 3000 extensión 31611 +52 33 1349 6920 (cell phone) daniel.sat@academicos.udg.mx, alternative email: satrathustra@gmail.com |

Reviewer 2 Report
The authors conduct a cohort study and aimed to determine the role of phase angle in the outcomes of head and neck cancer patients in a population with a high prevalence of overweight and obesity. The selected subjects were divided into normal and low phase angle (PA <4.42º and ≥4.42º.) groups and follow up for at least two years.
Comments:
1.
This information (Abstract): “The percentage of living patients (41.7%) is almost the same percentage of deceased subjects (43.1%; p=0.002)”
How to calculate the values of 41.7%(living patients) and 43.1% (dead subjects)?
Is it display in Tables, Figures or other?
2.
Figure 1 (image) is not clear; a high-resolution image should be provided.
KM plot should be showed the median with 95% confidence interval and the at-risk information(no. at risk) as well as effect size (e.g., Hazard ratio (95% CI))
e.g.,
The median overall survival was 38.6 months (95% confidence interval [CI], 34.5 to 41.8) in the osimertinib group and 31.8 months (95% CI, 26.6 to 36.0) in the comparator group (hazard ratio for death, 0.80; 95.05% CI, 0.64 to 1.00; P=0.046, Figure 1).
January 2, 2020 N Engl J Med 2020; 382:41-50
3.
Figure 1 a or Figure 1 b is a same finding, maybe the Figure 1 b should be selected (common).
4.
Table 4 (e.g., Month’s survival (Kaplan-Meier)) display the mean of survival (CI95% of the mean).
Lines 247-248, The median survival of patients with lower phase angle was 19.8 months versus 34.4 months in cases with normal phase angle, with a log-rank p-value <0.001 (Figure 1 a and b).
In general, KM plot is displaying the median survival (95% CI).
5.
How to calculated relative risk (RR) or hazard ratio (HR) by Cox proportional hazard model?
It is 42/60 divided 22/58 or 42/60 divided (22+10)/(58+21) for relative risk (RR)
or it is 42/60 patient-years of follow-up divided 22/58 patient-years of follow-up divided or 42/60 patient-years of follow-up divided (22+10)/(58+21) patient-years of follow-up for hazard ratio (HR).
6.
Characteristic of anthropometric and biochemical parameters(Tables 1 and 2) is significant between Phase angle <4.42º and ≥4.42º.
A multivariate analysis should be provided by Cox proportional hazard model, e.g., all variables with a P value of less than 0.05 in the univariate model were further entered into the multivariate analysis.
Author Response
COVER LETTER REVIEWER 2
Open Review
( ) I would not like to sign my review report
(x) I would like to sign my review report
English language and style
( ) Extensive editing of English language and style required
( ) Moderate English changes required
( ) English language and style are fine/minor spell check required
(x) I don't feel qualified to judge about the English language and style
|
Yes |
Can be improved |
Must be improved |
Not applicable |
|
|
Does the introduction provide sufficient background and include all relevant references? |
( ) |
( ) |
( ) |
(x) |
|
Are all the cited references relevant to the research? |
( ) |
( ) |
( ) |
(x) |
|
Is the research design appropriate? |
( ) |
( ) |
( ) |
(x) |
|
Are the methods adequately described? |
( ) |
(x) |
( ) |
( ) |
|
Are the results clearly presented? |
( ) |
(x) |
( ) |
( ) |
|
Are the conclusions supported by the results? |
( ) |
(x) |
( ) |
( ) |
Comments and Suggestions for Authors
The authors conduct a cohort study and aimed to determine the role of phase angle in the outcomes of head and neck cancer patients in a population with a high prevalence of overweight and obesity. The selected subjects were divided into normal and low phase angle (PA <4.42º and ≥4.42º.) groups and follow up for at least two years.
Comments:
Responses to reviewer 2 are highlighted in yellow
1.This information (Abstract): “The percentage of living patients (41.7%) is almost the same percentage of deceased subjects (43.1%; p=0.002)”
How to calculate the values of 41.7%(living patients) and 43.1% (dead subjects)?
In table 4 we modified the sum of the patient’s percentage, at the total patients column adding the 21 (15.1%) patients with lost follow-up, considering the live and death patients death at the moment in that we perform the analysis
Is it display in Tables, Figures or other?
This issue is described in table 4
- Figure 1 (image) is not clear; a high-resolution image should be provided
KM plot should be showed the median with 95% confidence interval and the at-risk information (no. at risk) as well as effect size (e.g., Hazard ratio (95% CI) added to the image for KM plot)
e.g.,
January 2, 2020 N Engl J Med 2020; 382:41-50
The median overall survival was 38.6 months (95% confidence interval [CI], 34.5 to 41.8) in the osimertinib group and 31.8 months (95% CI, 26.6 to 36.0) in the comparator group (hazard ratio for death, 0.80; 95.05% CI, 0.64 to 1.00; P=0.046, Figure 1).
According to the reviewer suggestion we added to the images the mean of survival and 95% confidence interval for the mean and the p value for log rank.
The mean survival was 19.8 months (95% confidence interval [CI], 15.6 to 24.1 months) in the phase angle<4.42 group and 34.4 months (95% CI, 29.6 to 39.2) in the comparison group (phase angle ³ 4.42: relative risk by Cox’s regression for death, RR=2.6, 95%CI=1.52 to 4.6; p<0.001, Figure 1).
When we used the sarcopenia/sarcopenic obesity to classify the study subjects the mean survival was 24 months (95%CI= 23.1 to 31 months) in without sarcopenia group the mean survival was 34.2 months (95%CI=28.1 to 40.3; p=0.033). Sarcopenia/sarcopenic-obesity relative risk by Cox’s regression for death, RR=1.9, 95%CI=1.04 to 3.7; p<0.036, in the Figure 1)
- Figure 1 a or Figure 1 b is a same finding, maybe the Figure 1 b should be selected (common); we selected only the survival KM plot.
We also eliminated the graphic describing the mortality rate (figure 1b and 1d corresponding to the v1. of the document. In the current version we have figure 1a and 1b).
4.Table 4 (e.g., Month’s survival (Kaplan-Meier)) display the mean of survival (CI95% of the mean).
Lines 247-248, The median survival of patients with lower phase angle was 19.8 months versus 34.4 months in cases with normal phase angle, with a log-rank p-value <0.001 (Figure 1 a and b).
In general, KM plot is displaying the median survival (95% CI).
We eliminated the information about survival and death rates and included that information in the survival plot images; table 4 contains only the information about the percentage of patients alive, dead and lost of follow-up and added the text according to the reviewer's suggestion
- How to calculated relative risk (RR) or hazard ratio (HR) by Cox proportional hazard model?
It is 42/60 divided 22/58 or 42/60 divided (22+10)/(58+21) for relative risk (RR)
or it is 42/60 patient-years of follow-up divided 22/58 patient-years of follow-up divided or 42/60 patient-years of follow-up divided (22+10)/(58+21) patient-years of follow-up for hazard ratio (HR).
We calculated the HR through automated Cox’s regression calculation using SPSS v28, introducing the survival months.
- Characteristic of anthropometric and biochemical parameters (Tables 1 and 2) is significant between Phase angle <4.42º and ≥4.42º.
A multivariate analysis should be provided by Cox proportional hazard model, e.g., all variables with a P value of less than 0.05 in the univariate model were further entered into the multivariate analysis.
We perform the analysis, but the variables were not significant, that is the reason we did not show the data.
We extended a bite the conclusion.
Submission Date
28 June 2022
Date of this review
13 Jul 2022 09:36:48
|
|
|
|
|
Dr. Luz Ma. Adriana Balderas Peña. M.D., PhD. Belisario Dominguez No. 1000. Colonia Independencia. Guadalajara, Jalisco, México, CP 44340 Telephone (office): +52 33 3668 3000 extension 31611 Telephone (mobile): +52 33 3115 7678 e-mail: luz.balderas@academicos.udg.mx, alternative e-mail: lmabp@yahoo.com.mx |
|
Dr. Daniel Sat-Muñoz MD, MBA 950 Sierra Mojada, Gate 7, Building C, 1st level, Colonia Independencia, Guadalajara, Jalisco 44340, México +52 33 3668 3000 extensión 31611 +52 33 1349 6920 (cell phone) daniel.sat@academicos.udg.mx, alternative email: satrathustra@gmail.com |
